# Indonesian Nursing Educators’ Experiences with Developing Student-Centered Learning Methods

**DOI:** 10.3390/nursrep15030081

**Published:** 2025-02-28

**Authors:** Vigdis Abrahamsen Grøndahl, Kirsti Lauvli Andersen, Ann Karin Helgesen, Asniar Asniar, Randi Martinsen, Riza Septiani, Dharina Baharuddin, Signe Rasch Woersaa, Anna Willman, Liv Berit Olsen

**Affiliations:** 1Faculty of Health, Welfare and Organisation, Østfold University College, P.O. Box 700, 1757 Halden, Norway; kirsti.l.andersen@hiof.no (K.L.A.); ann.k.helgesen@hiof.no (A.K.H.); liv.b.olsen@hiof.no (L.B.O.); 2Faculty of Nursing, University of Syiah Kuala, JL. Teuku Nyak Arief No. 441, Kopelma Darussalam, Kec. Syiah Kuala, Kota Banda Aceh, Aceh 23111, Indonesia; asniar@usk.ac.id; 3Department of Health and Nursing Sciences, Faculty of Social and Health Sciences, University of Inland Norway, P.O. Box 400, 2418 Elverum, Norway; randi.martinsen@inn.no; 4Department of Public Health, Faculty of Public Health, Universitas Muhammadiyah Aceh, Banda Aceh 23245, Indonesia; riza.septiani@unmuha.ac.id (R.S.); dharinabaharuddin@gmail.com (D.B.); 5Nursing, Institute for Health Sciences, University College South Denmark, Degnevej 16, 6705 Esbjerg Ø, Denmark; srwo@ucsyd.dk; 6Department of Health Sciences, Faculty of Health, Science, and Technology, Karlstad University, Universitetsgatan 2, 65188 Karlstad, Sweden; anna.willman@kau.se

**Keywords:** nursing education, pedagogic, qualitative content analysis, questionnaire, student-active methods

## Abstract

**Background/Objectives**: There is a rising global demand to educate independent and reflective nurses capable of addressing the evolving needs of healthcare systems and diverse populations. The Indonesian Nursing Act solidifies the professional status of nurses, although challenges persist including discrepancies between practice and educational standards. There is a need to increase the competence of Indonesian nurse educators and their ability to facilitate learning that can strengthen the nursing programs’ quality and improve the nurses’ ability to address various health issues in the population. The aim of the study was to describe the nurse educators’ experiences with developing student-active methods in nursing education in Indonesia. **Methods**: This study is part of an Erasmus+ project: Capacity Building in Nursing Education in Indonesia (CABNEI). A three-year educational program for nursing educators in Indonesia was developed. The current study used a qualitative, descriptive design with a questionnaire containing four open-ended questions focusing on the educators’ experiences with developing student-active methods and conditions affecting this process. Data were collected between January and November 2022 and analyzed using qualitative content analysis. **Results**: A total of 32 educators from two universities and two nursing schools in Indonesia participated. The analysis revealed the theme “A change of pedagogical approaches to nursing students’ learning” with three categories: “Setting the stage for pedagogical development in the nursing education”, “From correction to reflection”, and “Factors affecting the educators’ learning process”. **Conclusions**: The nursing educators’ experiences with developing student-active methods in Indonesia were positive. They reported incorporating additional pedagogical tools into their teaching methods and nursing programs. Management is vital in facilitating active-learning strategies. It is important to foster collaborative opportunities for educators and enhance their pedagogical skills to meet the healthcare system’s evolving needs and society’s demands for sustainable nursing expertise.

## 1. Introduction

The worldwide shortage of qualified healthcare professionals poses a significant threat to the health and welfare of individuals. The need for educating independent and reflective nurses capable of meeting the evolving healthcare system and the needs of the population is increasing [1]. The didactics and pedagogics within nursing education have evolved over time, and changes in nursing education can be seen as driven by the need to align the competencies of future healthcare professionals with the healthcare system demands and population needs [2]. The World Health Organization (WHO) recommends a pedagogical approach emphasizing active learning in nursing education [3]. In this study, the term pedagogy is understood as the approach to teaching, and as the theory and practice of learning. Nursing educators have a pivotal role in the pedagogical approach [4], and they need to have knowledge of the theories and principles of adult learning, curriculum implementation, communication, and collaboration to be able to implement active-learning methods in nursing education [5]. To achieve these competencies, educating teachers is of great importance. Skills in teaching and learning, clinical instruction, technology proficiency and evaluation, and particularly the need for ongoing support from colleagues, have been highlighted [6]. Mentoring and support from colleagues in nursing education are important and have been found to influence the nursing educators’ teaching style, moving the educators from a teacher-centered approach to a more student-centered focus [7]. Challenges for new nursing educators transitioning into teaching roles have also been explored. The results show that continuous learning about teaching was essential to the educators’ success, and that even experienced educators needed to continue enhancing their pedagogical competencies to educate competent and reflective nurses [8]. According to a study conducted by Pivač et al. [9], nursing educators reported that utilizing active-learning methods positively impacted the students’ communication and critical thinking skills. The study also emphasized the importance of providing appropriate facilities and ensuring that educators possessed adequate skills and knowledge on active learning and adult-learning principles to facilitate effective implementation.

The basis of active learning is reflection, a critical approach and questioning contexts, a willingness to learn, understanding, and interactions. Furthermore, active learning means being open to new perspectives by seeking new learning opportunities [10]. Active learning requires students to reflect professionally, which is crucial for their learning and educational development [11,12]. Implementing student-active learning methods can be approached in various ways like problem-based learning (PBL) and through simulation as a pedagogical method. In 2005, Jeffries [13] established a framework for the design, implementation, and evaluation of simulations to enhance teaching strategies in nursing education. A literature review [14] highlighted that simulation-based workshops or experiential training are widely employed approaches for improving skills and equipping educators to effectively utilize simulation as a pedagogical method. However, it is important to note that research also identifies challenges in revising the curriculum and implementing student-active learning methods. Institutional support is required to realize professional reflective education [4], and it is important to illuminate forces that support pedagogical changes and forces that resist pedagogical changes [15].

In Indonesia, aspiring nurses can choose between a three-year diploma or a four-year bachelor’s program. Challenges in the programs include poor perception, readiness, and the implementation of a results-based curriculum [16]. Edwards [17] examined the experiences of Indonesian nursing faculties involved in nursing education reform and concluded that curriculum changes can be complex and challenging. The participants’ opinions on factors influencing the curriculum should be acknowledged. The Indonesian Nursing Act solidifies the professional status of nurses, although challenges persist including discrepancies between practice and educational standards [18]. Since 2019, nursing education has aligned with the Indonesian National Qualification Framework. Educators are pivotal in maintaining high standards, requiring both subject matter expertise and pedagogical skills [19]. One study conducted in Indonesia emphasized the crucial role of leadership in facilitating student-centered learning activities [20]. Although student-active methods have been attempted in Indonesia, the transition to active learning seen in many Western countries has been underdeveloped [21,22]. This may be due to the traditional subordinate role of nurses in Indonesian healthcare, influencing nursing education [18].

The aim of the study was to describe the nurse educators’ experiences with developing student-active methods in nursing education in Indonesia.

## 2. Methods

### 2.1. Study Design

The study had a qualitative, descriptive design using a questionnaire with open-ended questions [23]. The manuscript adheres to the COREQ (COnsolidated criteria for REporting Qualitative research) Checklist (Appendix A) [24].

### 2.2. Setting and Participants

This study is part of an Erasmus+ project: Capacity Building in Nursing Education in Indonesia (CABNEI) including four Scandinavian and four Indonesian nursing educations (www.cabnei.com). The CABNEI-project aims to raise the competence of Indonesian nurse educators and their ability to facilitate learning that would strengthen the nursing programs’ quality and improve the nurses’ ability to address various health issues in the population. A three-year educational program for Indonesian nursing educators was developed, comprising three themes, starting in the first year with “nursing philosophy and theory”. The second year included “educational philosophy, views of learning, and teaching” and the last year of the project focused on “management of the nursing profession and the nursing role”. The current study focused on the Indonesian nursing educators’ experiences with the theme focusing on educational philosophy, views of learning and teaching in general, and developing student-active methods in particular.

A total of 32 educators from two universities at the bachelor’s level in nursing and two nursing schools at the diploma level in Indonesia participated in the educational program and answered the questionnaire. The educational program focusing on educational philosophy and views of learning and teaching consisted of four educational interventions each lasting five days. The teaching methods used during the educational interventions included lectures, simulations, role plays, discussions, reflection, and workshops. The participants from the four Indonesian educational institutions met to collaborate on themes tailored to the respective educational intervention. The specific themes included: (1) “Socio-cultural theories of learning” with the purpose of developing teaching methods that facilitate teacher–student dialog, cooperation in student groups, and systematic use of student evaluation for quality improvement work; (2) “Active learning such as problem-based and project-based learning” with the purpose of increasing the nursing educators’ competence to use methods that enable the students to play an active role in their own learning process; (3) “Learning, teaching, and supervision in clinical nursing” where the nurse educators practiced simulation as a method by carrying out training exercises, supervision, and reflection on the students’ learning process; and (4) “Testing, evaluating and discussing implementation of clinical methods in the curriculums” where the nurse educators developed methods for clinical studies and teaching together with nurse supervisors in clinical practice. In addition, they should discuss how to implement student-active methods and clinical methods in their curriculum. Due to the COVID-19 pandemic, the first two educational interventions were conducted online. The third intervention took place in Indonesia, and the final one took place in Norway.

Furthermore, two workshops were conducted to promote reflection on individual practices and to enhance the educators’ understanding of the themes covered in the educational intervention. The educators first developed an implementation strategy for academic and active-learning methods, and then conducted different pedagogical methods in nursing including working in groups and facilitating simulation. Each workshop lasted five days. The two workshops were facilitated by the local change agents who were among the 32 educators in the project. The first workshop took place in Indonesia between the first and second educational intervention and included the educators from the four Indonesian educational institutions. Project members from Scandinavia met the educators online three times during the week for supervision and discussion. During the second workshop, change agents from Indonesia and project members from Scandinavia met in person in Norway.

The participants were mostly women (29 out of 32 participants), with a mean age of 44 years (26–56 years) and different educational levels (Table 1). Their teaching experience varied from 2 to 35 years, with an average of 18 years.

### 2.3. Data Collection

A questionnaire was developed and discussed among three of the authors (VAG, KLA, and AKH), all with high formal pedagogical competence and several years of experiences in teaching and supervising in nursing and healthcare at the bachelor-, master- and PhD level. In total, four open-ended questions were formulated and included the following questions: “What has contributed to your learning?”, “What has been useful for you?”, “How have you worked on your own learning process?”, and “Please give suggestions for improvement”. In addition, data on the participants’ gender and educational level were collected. Data were collected between January and November 2022. The participants were invited to answer the questionnaire after each educational intervention, a total of four times. The participants received the paper-based questionnaire by one of the authors (AA) and decided when and where to answer the questions. The participants who volunteered to answer the questionnaire did so within one week after completing the educational program and gave their written answers in their native language (Bahasa Indonesia). A total of 107 completed questionnaires were collected following each educational intervention, with 32, 21, 30, and 24 questionnaires acquired, respectively. One of the native-speaking authors was responsible for collecting both the questionnaires and translating the responses into English (AA).

### 2.4. Ethical Considerations and Approval

This study is part of an Erasmus+ project: Capacity Building in Nursing Education in Indonesia (CABNEI) (No. 619083-EPP-1-2020-1-NO-EPPKA2-CBHE-JP). All deans of the participating educational institutions consented to their institution’s participation in the project, and to promote and play an active role in dissemination of the results from the different studies in the project by signing the Partnership Agreement. This study was conducted in accordance with the ethical principles of the Declaration of Helsinki [25]. The 32 educators who participated in the educational program were informed orally about the study, that answering the questionnaire was voluntary, and that they gave their informed consent by answering and returning the questionnaire. Finally, for the educational intervention, the participants received the questionnaire and were asked to complete it within one week. The questionnaire did not contain any personal information, only anonymous data. According to ACT 2018-06-15-38: Act on the Processing of Personal Data (the Personal Data Act) [26], the study shall not be notified to the data protection officer. Background information about the 32 participants was therefore collected from the heads of their respective educational institutions after the educational interventions were completed.

### 2.5. Analysis of Data

The data were analyzed using Graneheim and Lundman’s qualitative content analysis [27]. The text was initially read several times by two of the authors (VAG and KLA) to familiarize themselves with the content and to obtain an overview of the text. The same two authors individually read all 107 responses and identified meaning units in line with the aim of the study. The identified meaning units were condensed, abstracted, and labeled with a code. The codes were compared and sorted into one theme with three categories based on similarities and differences. The two authors systematically revisited and questioned the codes, categories, and themes to ensure that they were firmly grounded in the data rather than influenced by the researcher’s assumptions, thereby enhancing the credibility and validity of the results by minimizing the impact of the researcher’s preconceptions. Changes were made before all co-authors discussed the revised categories until consensus was reached.

## 3. Results

### 3.1. A Change of Pedagogical Approaches to Nursing Students’ Learning

The analysis revealed the theme “A change of pedagogical approaches to nursing students’ learning” including the following three categories: “Setting the stage for pedagogical development in the nursing education”, “From correction to reflection”, and “Factors affecting the educators’ learning process”. The theme and categories are presented in Table 2.

#### 3.1.1. Setting the Stage for Pedagogical Development in the Nursing Education

The participants expressed that they enhanced their competencies by gaining insights into new learning theories and pedagogical methods and engaging in practical exercises during the educational interventions and workshops. The importance of group work involving academic discussions and reflections on the impact of different educational institutions’ perspectives on learning was highlighted, fostering a motivation to advance the pedagogical development of the curriculum further. One of the participants stated it as follows:


*“I feel that the existing curriculum design needs to be improved because there are still many learning methods (from CABNEI) that can be improved...”*
(Participant 7)

The knowledge of active learning like problem-based learning (PBL) that the participants gained through the workshops gave them confidence that they could implement new pedagogical methods in both nursing and public health education at their institution. Two of the participants said:


*“Implementation of PBL, integration of syllabus with PBL methods, listening to the experience of foreign universities, adopting the concept of active learning in the teaching and learning process in nursing, have contributed to my learning.”*
(Participant 9)


*“... my knowledge and skills have increased so that I can implement them in the teaching process.”*
(Participant 10)

Throughout the workshops, the participants from the four Indonesian universities collaborated closely. Knowledge and experience from nursing in general and nursing education in particular were shared. The participants expressed that it was useful to share experiences with partners from both Scandinavia and Indonesia:


*“Sharing experiences and assessing what has been done and things that need to be improved in our education.”*
(Participant 25)

The participants expressed a desire to continue networking both locally in Indonesia and with their Scandinavian partners after the project ended as they experienced that it was one way of developing quality in nursing education. One quote from a participant describes the networking:


*“It has been useful for me getting to know our Scandinavian partners (personally and professionally) and increased my bond with other Indonesian team members. Now that I have finally met and can be together with the other team members from other institutions, I am more confident in communicating, interacting, and reaching out to them.”*
(Participant 23)

#### 3.1.2. From Correction to Reflection

The participants expressed that by being introduced to social learning theory and the didactic relationship model, they had gained an expanded theoretical knowledge of learning theories and learning models as the foundation for different approaches to teaching in nursing education. Consequently, they indicated acquiring additional pedagogical strategies for designing courses and lectures. One participant said:


*“I can analyze learning by using several models/concepts and solve problems in education.”*
(Participant 4)

Participants also expressed that they had begun to apply the pedagogical theories when planning their teaching:


*“I am designing teaching using social learning theory.”*
(Participant 10)

The participants experienced that by altering the pedagogical approach from simply telling students what was right or wrong to encourage them to reflect on and discuss their actions, the educators shifted from a punitive method of correction to one that fosters learning through self-reflection. One participant described this shift as a change in mindset:


*“This workshop is beneficial because it changes the mindset of how a lecturer develops pedagogic skills using the correct teaching methods.”*
(Participant 9)

Student-active methods and the use of active-learning methods were only partially visible in the Indonesian curriculums. The participants described that they had always used a master–apprentice approach when teaching nursing. During the educational interventions, they expressed a wish to make the students more active during the learning process. They also expressed that it was useful to learn from colleagues in other universities who participated in the workshops.


*“The benefits are many, increasing knowledge and sharing each other’s experience about active learning.”*
(Participant 19)

Participants indicated that they gained increased competence in simulation as a pedagogical method that contributed to learning by reflection. They emphasized that both the theoretical foundation regarding the development of relevant scenarios based on authentic situations and the simulation phases were important. They described an increased understanding and knowledge of developing scenarios, preparing for all phases in the simulation, and carrying out a simulation including how the simulation could be implemented as part of the nursing curriculum. The importance of structure and systematic, in addition to the role of the facilitator in simulation, were highlighted. One participant expressed it as follows:


*“I have a better understanding of scenario development, simulations, and clinical methods in curriculum development. I understand more about simulations and preparing for simulations and doing them.”*
(Participant 9)

Several participants expressed a changed perspective on their role as teachers. They discovered that facilitating students in learning situations was more critical in the simulation than telling students what the right way was when their goal was to increase the nursing students’ critical skills. The debriefing phase was also highlighted. This contributed to the participants’ learning. Two of the participants expressed it as follows:


*“I have learned how to do simulation correctly and not interrupt students, because the lecturer is a facilitator.”*
(Participant 4)


*“... now my knowledge has improved that simulation is a learning process creating actual situations students will face. Simulation is an effective learning method to increase students/ skills and critical thinking.”*
(Participant 27)

#### 3.1.3. Factors Affecting the Educators’ Learning Process

Various factors related to cultural context, physical environment, and working conditions influenced the educators’ opportunity for professional learning. The value of participating in a group with cultural diversities was expressed. Both learned from members of other educational institutions in Indonesia and from members from Scandinavian educational institutions.


*“I get a lot of new knowledge from experts, learn different cultures from our daily culture, increase the experience of working in groups and learn from each other from sharing experiences.”*
(Participant 1)

The language used in the workshops was English, and none of the participants used English as their native language. In addition, there was a variation in language skills, and for some of the participants, the barriers might have affected the learning outcomes. Various solutions were proposed to reduce these barriers such as translating the PowerPoint presentation into Bahasa and distributing them to the participants well in advance of the lectures. One of the participants expressed it as follows:


*“During the discussion, I saw that colleagues didn’t understand the explanation well. I don’t know if it might be a lack of focus or a language barrier. If possible, it can be translated directly after each presenter has explained each slide.”*
(Participant 15)

In addition to language barriers, the participants experienced some challenges due to the heavy workloads and troublesome facilities. The participants had to manage their duties as teachers parallel to taking part in the workshops, which resulted in participants coming and going during the gatherings. This was expressed by some of the participants as somewhat annoying. Two of the participants said:


*“Workshop activities have been going well and are very useful; it is necessary to improve the discipline of participants’ attendance.”*
(Participant 12)


*“It needs commitment from the participants to be able to follow the workshop from the beginning to the end of the activity.”*
(Participant 21)

The participants described challenges concerning that the project and the workshops started in parallel with the outbreak of the COVID-19 pandemic. Due to the lack of possibility to meet face to face, it was of utmost importance that the participants had well-functioning technical equipment for remote interactions during the workshops. The participants underscored that they all had to step up their knowledge and use of digital collaboration. In addition, the Internet connection was at times unstable, and as one participant expressed:


*“There are also occasional exits from the network and some for almost 30 min; the cell phone has an error. The Internet connection is problematic, it goes in and out of digital meetings.”*
(Participant 6)

Despite challenges with the Internet, the participants also experienced that the digital workshops worked well, but that they missed the physical meeting.


*“So far, it’s been good, but I want to meet in person and not just in Zoom.”*
(Participant 9)

The lack of a physical environment adapted to teaching and discussion during the workshops was another obstacle that the participants expressed as a condition that negatively affected their learning process during the first year of the project. One participant underscored the importance of not being distracted during the digital encounters:


*“It will be better if the workshop is held in a comfortable atmosphere with beautiful views so it will not be distracted by other activities.”*
(Participant 9)

## 4. Discussion

This study investigated the nursing educators’ experiences with developing student-active methods in nursing education in Indonesia. The data analysis revealed an overarching theme: “A change of pedagogical approaches to nursing students’ learning”. This can be interpreted as positive experiences, with educators reporting that they have incorporated additional pedagogical tools into the nursing education program. The change was categorized by setting the stage for pedagogical development in nursing education, shifting from correction to reflection, and identifying factors affecting the educators’ learning process.

The educators in our study discovered that their curricula needed to be revised and developed to include active learning and simulation, which they found essential for fostering a more student-centered and active pedagogical approach. Several studies show that institutional support is required when developing curriculums [4,14,15]. None of the educators expressed a lack of institutional support but perceived their own role as important for changing the pedagogical approach, and to be able to do so, they needed to develop their knowledge in pedagogy. This is consistent with findings from a previous study [4]. There is an expectation that educational institutions regularly review and update their curricula to uphold the quality of education and ensure their alignment with societal needs [1]. Curriculum development based on educational theory and evidence-based teaching practices focusing on active learning is among the World Health Organization’s core competencies in nursing education [3,5]. By being introduced to social learning theory and didactics, the educators in our study expressed that they had gained insight into student-active learning, and that active learning was vital for educating a reflective and independent nurse. The educators acknowledged that the nursing students were individual people, and they became more aware of the interaction between educators and nursing students. Educators are expected to create a learning environment that positively impact the nurse students’ ability to reflect and socialize with other students [3]. However, results from a previous Indonesian study found that most nursing schools focused too much on curriculum development, teaching processes, and assessment in comparison to the impact that a more student-centered pedagogic had on the students’ learning and how to design the learning environment [20]. Shehnaz and Sreedharan [28] underscored the importance of how the educational environment impacted the nursing students’ learning experiences. Educators are responsible for designing a learning environment that fosters effective learning experiences. To develop a curriculum that meets the desired learning outcomes, they must have a deep understanding of how students learn.

The educators in our study varied in age, level of education, and teaching experience. However, they all expressed the usefulness of sharing experiences and knowledge with colleagues from other nursing educations in Indonesia and Scandinavia. Moreover, they highlighted the opportunity to reflect on the perspectives of the different educational institutions on learning and expressed a wish to continue collaborating when the project ended. This is in line with results from a previous study focusing on building pedagogical and professional competence among educators in Indonesia after the tsunami [29]. The educators were encouraged by collaboration and teamwork, and by more active students. They felt more independent as teachers and were inspired to develop further competencies [29]. The World Health Organization [5] has also emphasized collaboration and communication between educators in nursing by including those values among the nurse educators’ core competencies and highlighting both collaboration between education and clinical practice and intercultural and interdisciplinary partnerships. Previous research showed that mentoring and ongoing support from colleagues, especially from experienced colleagues, are important to develop the educators’ teaching competence [6,7]. In addition, an integrative review explored the challenges for new nurse educators transitioning into teaching roles and identified factors aiding in this transition, emphasizing that the educator learning about teaching is crucial for success, and that experienced educators need to continue working on and enhancing their competencies in pedagogy to educate competent and reflective nurses [8]. There is, however, an increasing shortage of qualified nursing educators globally, which may have a negative impact on the need for mentoring and the support of new educators in nursing [30], which again, may affect the quality of nursing education negatively. Despite the challenges and potential obstacles, it is crucial for teachers to recognize the importance of providing optimal support to one another in developing teaching competencies and motivating each other within the framework in which they operate.

The educators in our study experienced that they acquired new learning theories and gained a broader understanding of different pedagogical approaches and methods focusing on active learning in nursing education: project work, group work, and simulation. They expressed a desire to implement these methods at their own educational institutions to strengthen the students’ critical thinking skills. This is in line with results from a review that states that simulation-based workshops increase the possibility for educators to utilize simulation as a pedagogical method in their own teaching [14]. Nurses in Indonesia are characterized by assisting the doctor, and there is little emphasis on the nurse’s ability to work independently [31]. Managing and motivating nursing students are challenging, and addressing diverse learning needs is essential for teaching competence [6]. Having competence in pedagogical methods that enable educators to motivate students toward becoming independent professional nurses with reflection skills is crucial. Knowledge of the theories and principles of adult learning and active learning is essential for nursing educators to ensure high-quality education in nursing. The educators in our study expressed that by participating in the educational program, they had changed their way of interacting with students from correcting them to motivating them to reflect on their own actions. These results are supported by a systematic review that stated that guided reflection strategies in nursing education change how students learn and is a vital part of simulation-based education that enhances the participants’ knowledge and reduces the gap between education and clinical practice [32]. In addition, facilitating critical thinking education improves the students’ problem-solving skills [33] and fosters professional growth and learning by encouraging the students to question their own knowledge [34]. It has also been found that it is important to include collaboration, student-centeredness, and active-learning methods when facilitating the development of critical thinking skills among nurse students [9,34]. This corresponds with what the educators in our study experienced.

The educators expressed that some conditions influenced their opportunities for professional learning. The learning environment played a crucial role. Troublesome facilities, such as a lack of adequate physical space for discussion during workshops, tended to inhibit interactive learning experiences. Unstable Internet and poorly functioning technical equipment created barriers to online learning and virtual collaboration and obstructed smooth communication. Language barriers sometimes made it difficult for the educators to fully engage with the material and contribute meaningfully to discussions. These challenges are described in previous research [9]. The lack of educational resources was found to be an obstacle for implementing the nurse educators’ core competencies globally, and particularly in low- and middle-income countries. To address this challenge, a platform for open education resources with freely accessible curriculum materials and remote monitoring support was established by Nurse International [30]. Providing appropriate facilities must be emphasized to ensure the best possible environment for learning. The management and leadership of educational institutions must facilitate conditions for educators to be able to initiate and use student-active learning methods in the training of future nurses. On the other hand, a study from Tajikistan, assessing the effect of the nursing education reform on the educational environment, found that even if the educational environment had improved, there was still need for targeted efforts to improve the didactical competencies of educators to increase competency-based learning and skills training [35]. This was also recognized in a South African study where nursing educators were resistant to change, as they thought that the revised curriculum, with an innovative, student-centered approach, did not encourage higher cognitive learning. In addition, scarce resources were described to influence the teaching and learning environment and the ability to make use of pedagogical methods that promoted the students’ skills in clinical reasoning [36]. The students’ readiness for the simulation, the need for team building among the educators, a lack of adequate resources, and thoughtful integration of the simulation into nursing curricula were found to be some obstacles for making use of student-active methods [37]. At the same time, the educators perceived that simulation as an active way of learning motivated the students’ learning process [37]. Educational conditions like a relaxed learning climate have been found to be important for facilitating the development of critical thinking, and teaching strategies where both educators and students are active in supporting such milieu should be targeted [36].

This study had some limitations. Due to the COVID-19 pandemic, two educational interventions had to be conducted online. Moreover, there were challenges with poor Internet coverage, which may have affected the quality of the intervention. This was addressed by dividing the participants into smaller groups, each of which included a project participant from Scandinavia, who led the reflections related to the content of the specific intervention. The data collection was limited to 32 educators, which may have restricted the amount of data obtained. Including more participants could have provided a greater depth of information. A strength of the study was that the participants answered the open-ended questions after attending the educational interventions, which gave a total of 107 answers. In addition, the participants worked in four different educational institutions, and their levels of education differed, providing rich material. The answers to the open-ended questions were written in Bahasa Indonesia, and one of the Indonesian authors was responsible for translating the answers into English. Some words may have been misunderstood, and this may have influenced the results. As none of the authors were native English speakers, the translation of the answers was discussed among all the authors to clarify possible misunderstandings. The data were rigorously analyzed following Graneheim and Lundman’s phases [27] to improve transparency. Two of the non-Indonesian authors conducted the first analysis and organized the results into theme and categories. The analysis and interpretation could have been influenced by the two authors’ cultural background. To increase cultural sensitivity and understanding of the data and prevent possible misunderstandings, all authors had regular meetings to discuss and reflect on these together.

## 5. Conclusions

The nursing educators’ experiences with developing student-active methods in nursing education in Indonesia were positive. They reported incorporating additional pedagogical tools into their teaching methods and nursing programs. The role of management is pivotal in facilitating the implementation of active-learning strategies within curricula. Shifting the educational focus from correction to reflection has the potential to foster more reflection and greater independence among nurses in Indonesia’s healthcare system, thereby enhancing the quality of patient care. Efforts should be made to foster opportunities for educators to engage in collaborative initiatives across educational institutions both nationally and internationally. Additionally, it is essential for educators to continuously enhance their pedagogical competencies to meet the evolving needs of the healthcare system and society’s demands for sustainable nursing expertise in the future. Further studies are needed to provide recommendations on how Indonesia’s nursing curriculum can effectively integrate active-learning methods.

## Figures and Tables

**Table 1 nursrep-15-00081-t001:** Overview of the participants’ educational level (N = 32).

Level of Education	N = 32
Doctoral degree in nursing or public health	8
Master in nursing science, health science or public health	18
Bachelor or Diploma in nursing	6

**Table 2 nursrep-15-00081-t002:** Overview of the theme and the categories.

A Change of Pedagogical Approaches to Nursing Students’ Learning		
Setting the stage for pedagogical development in the nursing education	From correction to reflection	Factors affecting the educators’ learning process

## Data Availability

The data are available upon request to the first author to maintain the confidentiality and anonymity of the educators taking part in the study.

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
