# Peer review of "Indonesian Nursing Educators’ Experiences with Developing Student-Centered Learning Methods"

_nursrep, 2025, doi:10.3390/nursrep15030081_

Round 1

Reviewer 1 Report

Comments and Suggestions for Authors

Dear authors, 

I thank you for allowing me to review your manuscript. I find the topic presented interesting. However, I also feel that some corrections could improve the comprehensibility of the manuscript. 

____________________________________

ABSTRACT: 

Background/Objectives: 

Line 19-25: The first sentence is disjointed from everything else; the fact that the study is part of an ERASMUS+ project does not help to understand or give background on what was done and why. In this part, the study's aim is not sufficiently clear. I suggest a reworking that begins by emphasizing how the training of nurse educators is crucial in the global context, then focusing on the local (Indonesian) aspects. Finally, make the purpose explicit well. 

Methods: 

Line 29-30: “Four open-ended questions focusing on the educators‘ learning process and conditions affecting this process.” but in the purpose (line 112-113), the authors state, “The aim of the study was to describe nurse educators’ experiences with developing student-active methods in nursing education in Indonesia.” Written in this way, the aim and instrument used to achieve it are not concordant. 

I suggest the authors revise both depending on what may have been investigated by the four open-ended questions. 

Results: 

The authors correctly reported that the response sample consisted of 32 nurse educators. However, these completed 107 questionnaires. In my opinion, this should be specified. 

The nursing educators are stated to be from two universities and two nursing schools in Indonesia (total: 4 learning campuses). This same sentence is echoed in Line 122. However, both Lines 126 and 222 mention “participants from 8 universities.”  Is it four, or is it 8? The authors must explain this point better and harmonize it in the manuscript. 

Conclusions: 

Line 37: “They increased their didactic competence.” There is no objectivity in this sentence. The authors may write that the teachers reported, in their opinion, that they had improved their teaching competence, but stating that these were not objectively measured beforehand and that measuring them was in no way the purpose of the study; this statement is incorrect and should be changed, both in the abstract and in the manuscript. 

OTHER INDICATIONS: 

Line 101-111: this part should go under "methods" section.

Throughout the manuscript, pedagogy/pedagogical skills are mentioned many times. However, I do not see cited or reported theories of learning that should be central to an article of this kind. For example, lines 79-81: a literature review is used to talk about simulation-based learning or experiential learning (cited as experiential training), but neither Kolb's theory (Experiential Learning, 1984), which is fundamental to modern nursing education) nor is Jeffries' (more recent...) simulation mentioned.

Let me be clear: I am not asking the authors to cite either one (with which I would have no conflict of interest anyway); however, on a general level, if one does not compare this with a solid theoretical basis, the background of this article is highly fragile. 

Another issue the term “Pedagogy" / “pedagogical competence”; it would be appropriate for the authors to clarify in the first parts of the article that they mean the term in its most generic meaning possible since a relatively widespread understanding of pedagogy is that it concerns the training of children and adolescents and that andragogy focuses, instead, on adults. In this meaning, the term would not be correct and could lead to misunderstandings. 

Finally, since this is a qualitative study, it would be appropriate to clarify whether (and if so, how) the authors have addressed bracketing; if it has not been addressed, it should be specified within the limits, necessarily specifying what this might entail. 

Comments on the Quality of English Language

English could be improved but is comprehensible.

Reviewer 2 Report

Comments and Suggestions for Authors

The use of active learning methods in nursing education and their effects are frequently studied. This study differs from other studies with its sample and study method. The general structure of the article complies with academic writing rules, but there are some points that need to be corrected.

1. Did the authors use any programs to evaluate qualitative data?

2. The themes and subthemes determined in the content analysis can be more understandable if given in a diagram.

3. The workshop's content mentioned in the method section should be given in more detail. Additional information should be provided regarding the purpose of these studies, their place in the research, and the workshop's content.

4. Which courses do the participants teach in nursing education? What are their areas of expertise? Have they used any active learning techniques before? Answers to these questions can also be added.

5. There is a lot of space between the participant statements and comments in the Results section; this situation can be corrected.

6. The reference indication in line 404 should be corrected.

7. Participant characteristics can be given in a table.

Author Response

Please see attachement.
